# 'Online' integration of sensory and fear memories in the rat medial temporal lobe

Francesca S Wong, R Fred Westbrook, Nathan M Holmes*

School of Psychology, University of New South Wales, Sydney, Australia

**Abstract** How does a stimulus never associated with danger become frightening? The present study addressed this question using a sensory preconditioning task with rats. In this task, rats integrate a sound-light memory formed in stage 1 with a light-danger memory formed in stage 2, as they show fear when tested with the sound in stage 3. Here we show that this integration occurs 'online' during stage 2: when activity in the region that consolidated the sound-light memory (perirhinal cortex) was inhibited during formation of the light-danger memory, rats no longer showed fear when tested with the sound but continued to fear the light. Thus, fear that accrues to a stimulus paired with danger simultaneously spreads to its past associates, thereby roping those associates into a fear memory network.

DOI: https://doi.org/10.7554/eLife.47085.001

## Introduction

Memory is the means by which the past connects with the present to guide our future interactions with the environment. One of the ways in which memory achieves this remarkable feat is by integrating the sensory and emotional elements of experiences that are separated in time but share common elements. For example, suppose that while hiking through a rainforest in tropical Australia you hear an (unfamiliar) deep booming sound and then see the (unfamiliar) bird-like animal who is making that sound. The next day, you see a poster showing a photograph of the animal (a cassowary) with a heading 'extremely dangerous.' Finally, when again hiking through the rainforest you hear the booming sound and feel apprehensive, even frightened.

This hypothetical scenario illustrates how the sound-animal memory formed at one point in time and the animal-danger memory formed at a second point were integrated via their common, animal, element to generate defensive/fear reactions to the sound. This type of integration could occur in two ways (*Shohamy and Daw, 2015*; *Schlichting and Preston, 2015*). The first is through formation of indirect associations between past sensory experiences and emotional events as they are encountered (*Hall, 1996*; *Holland, 1981*; *Konorski, 1967*; *Ward-Robinson and Hall, 1999*): that is during formation of the animal-danger memory, the past sound-animal memory is activated and integrated with the emotional memory content (what might be called 'online' integration). The second is through retrieval and chaining of memories at choice points (*Rizley and Rescorla, 1972*; *Sadacca et al., 2018*; *Sharpe et al., 2017*): that is re-exposure to the sound activates the sound-animal memory, which in turn, activates the animal-danger memory, resulting in expression of defensive/fear reactions to the sound ('memory-chaining').

At present, very little is known about when/how such memories are integrated in the brain to generate action. The present study addressed these questions using an animal model, sensory preconditioning in rats. We specifically examined whether sensory and emotional memories, which are consolidated in distinct regions of the medial temporal lobe (MTL) (*Holmes et al., 2013*; *Holmes et al., 2018*; *Parkes and Westbrook, 2010*), are integrated online when new memories are formed, or through a memory-chaining mechanism when actions are required. The protocol consisted in three stages that were separated by 24 hr. In stage 1, rats are exposed to pairings of two

*For correspondence:
n.holmes@unsw.edu.au

**Competing interests:** The authors declare that no competing interests exist.

relatively innocuous stimuli (e.g., a sound followed by a light), and in stage 2 they are exposed to pairings of the light and footshock. Finally, in stage 3, rats are tested with the sound and then with the light. Rats typically exhibit defensive reactions (freezing) when tested with the preconditioned sound, even though it had never been paired with danger, and when tested with the conditioned light, that had been paired with danger.

Control conditions confirm that freezing to the sound in this protocol requires formation of sound-light and light-shock memories in training (it is not due to generalization of fear), and previous work in our laboratory has shown that the substrates of these memories are doubly dissociable in the MTL (*Holmes et al., 2013*; *Holmes et al., 2018*). The sound-light memory that forms in stage 1 requires neuronal activity, including activation of NMDA receptors, in the most anterior region of the parahippocampal cortex, the perirhinal cortex (PRh), but not the basolateral amygdala complex (BLA) (*Holmes et al., 2013*; *Parkes and Westbrook, 2010*); whereas the light-shock memory that forms in stage 2 requires neuronal activity, including activation of NMDA receptors, in the BLA, but not the PRh (*Parkes and Westbrook, 2010*; *Wilensky et al., 2006*). The major question of interest is *when* the sound-light and light-shock memories are integrated to generate freezing responses to the test presentations of the sound.

## Results

### Experiment 1: Demonstration of sensory preconditioned fear to the sound

Experiment 1 had two aims. The first was to show that rats exposed to sound-light pairings in stage 1 and light-shock pairings in stage 2 exhibit fear when tested with the sound alone in stage 3. The second was to show that this fear of the sound is conditional on pairings of the relevant stimuli in each stage of training: that it is due to sensory preconditioning, rather than generalization of fear from the light to the sound. Rats in the group of interest, Group PP, were trained in the manner just-described (*Figure 1A*). Rats in the control groups were exposed to unpaired presentations of the sound and light in stage 1, followed by light-shock pairings in stage 2 (Group UP); or to sound-light pairings in stage 1, followed by unpaired presentations of the light and shock in stage 2 (Group PU). All rats were then tested for freezing to the sound and light alone in stage 3.

During conditioning in stage 2 and testing in stage 3, the baseline levels of freezing were low (<15%) and did not differ between the groups (largest $F$ = 1.841; p = 0.182). Conditioning of the light was successful in Groups PP and UP in stage 2. The mean (± SEM) levels of freezing to the light on its final pairing with shock were 79 ± 7.7% in Group PP, 69 ± 8.9% in Group UP, and were 39 ± 7.6% on the final presentation of the light in Group PU. Groups PP and UP did not differ in their overall levels of freezing to the light ($F$ = 0.058; p = 0.811) but froze at significantly higher levels than Group PU ($F_{(1,42)}$ = 42.641; p < 0.0001; $d$ = 0.91; 95% CI: [22.522, 42.670]). There was a significant linear increase in levels of freezing across the light-shock pairings ($F_{(1,42)}$ = 143.188; p < 0.0001; 95% CI: [42.780, 60.137]). The rate of increase did not differ between Groups PP and UP ($F$ = 0.765; p = 0.387), but this increase was significantly different from that in Group PU ($F_{(1,42)}$ = 8.793; p = 0.005; 95% CI: [8.484, 44.641]).

*Figure 1B* shows the mean (± SEM) test levels of freezing averaged across the eight sound trials (left panel), and shows the mean (± SEM) test levels of freezing averaged across the eight light trials (right panel; see also *Figure 1—source data*). Rats exposed to pairings of the sound-light in stage 1 and to light-shock pairings in stage 2 (Group PP) froze significantly more to the sound than rats that received pairings in stage 1 but not in stage 2 (Group PU) and those that received pairings in stage 2 but not in stage 1 (Group UP) ($F_{(1,42)}$ = 20.684; p < 0.0001; $d$ = 0.91; 95% CI: [7.063, 18.330]). Rats in these two latter groups did not differ from each other in their levels of freezing ($F$ = 0.043; p = 0.837). The analysis also confirmed what is clear from inspection of the light test. Rats that received light-shock pairings (Groups PP and UP) did not differ in their levels of freezing from each other ($F$ = 3.988; p = 0.052) but froze significantly more to the light than rats that received unpaired light-shock presentations (UP) ($F_{(1,42)}$ = 66.477; p < 0.0001; $d$ = 2.06; 95% CI: [37.059, 61.439]). Thus, freezing to the sound in this protocol is not due to generalization of fear, but rather, requires formation of sound-light and light-shock memories in training.

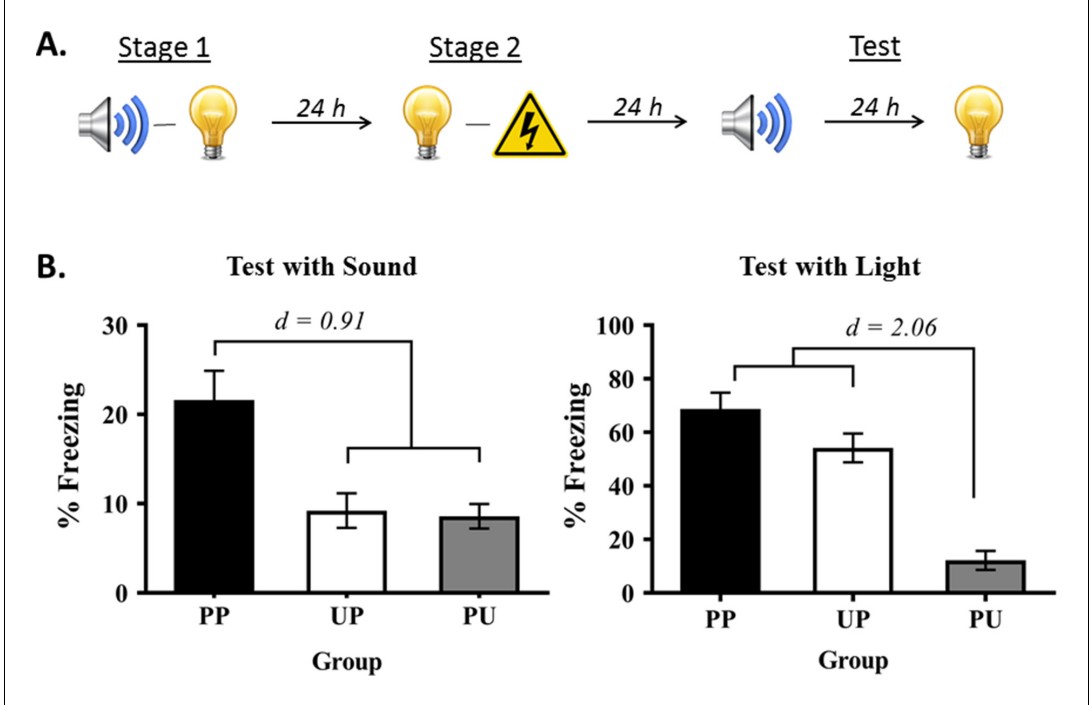

**Figure 1.** Demonstration of sensory preconditioned fear. (**A**) Schematic of the behavioral protocol used for the group of interest (Group PP) in Experiment 1, and in all subsequent experiments (Group PP, *n* = 13; Group PU, *n* = 16; and Group UP, *n* = 16). It is important to note that the sound and light were fully counterbalanced across the preconditioned and conditioned stimulus identities in this experiment and all subsequent experiments. For convenience of explanation, the designs are described with reference to one half of the counterbalancing. (**B**) Percentage freezing to the preconditioned sound (left panel) and to the conditioned light (right panel), averaged across the eight trials of their respective tests. It is also important to note that the level of freezing to the preconditioned sound is never as high as the level of freezing to the directly conditioned light (see *Holmes et al., 2013*; *Holmes et al., 2018*; *Parkes and Westbrook, 2010*). For this reason, we avoid drawing direct comparisons between the preconditioned sound and directly conditioned light, and use y-axis scales that most effectively show the key findings. Data shown are means ± SEM. They were analyzed using sets of planned orthogonal contrasts (*Hays, 1963*). For test levels of freezing to the sound, the first contrast compared Group PP versus the weighted average of Groups PU and UP; and the second contrast compared Group PU versus Group UP. For test levels of freezing to the light, the first contrast compared Group PU (no light-shock pairings) versus the weighted average of Groups UP and PP (both received light-shock pairings); and the second contrast compared Group UP versus Group PP. Cohen's d (*d*) is shown for statistically significant results. For raw data, see the *Figure 1—source data 1*.

DOI: https://doi.org/10.7554/eLife.47085.002

The following source data is available for figure 1:

**Source data 1.** Demonstration of sensory preconditioned fear.

DOI: https://doi.org/10.7554/eLife.47085.003

## Experiment 2: Consolidation of the sound-light association formed in stage 1 requires de novo protein synthesis in the PRh

In Experiment 2, we used micro-infusions of a protein synthesis inhibitor, cycloheximide, to confirm that the sound-light memory that forms in stage 1 is consolidated in the PRh (*Figure 2A*). Protein synthesis is thought to be critical for the structural changes that stabilize new information in memory and/or permit successful retrieval of stored information (for reviews, see *Davis and Squire, 1984*; *Hernandez and Abel, 2008*). Therefore, if consolidation of the sound-light memory requires protein synthesis in the PRh, it would be disrupted by an infusion of cycloheximide, and this disruption would be reflected in less freezing to the sensory preconditioned sound. Rats were exposed to sound-light pairings in stage 1, and immediately after this session, infused with either cycloheximide (Group CHX) or vehicle alone (Group VEH) into the PRh. All rats were then exposed to pairings of the light and shock in stage 2, and finally, tested with presentations of the sound and light in stage 3.

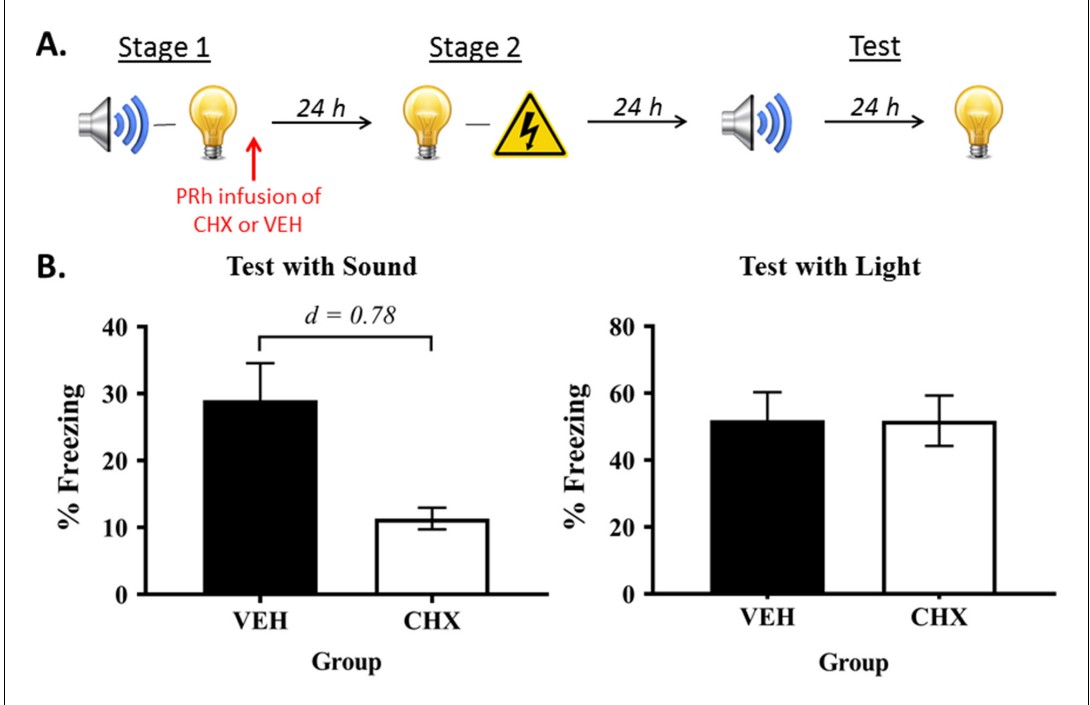

**Figure 2.** Protein synthesis in the PRh is required for consolidation of a sensory sound-light memory. (**A**) Schematic of the behavioral procedure for Experiment 2 (Group VEH, n = 8; and Group CHX, n = 10). The red arrow indicates infusions occurred immediately after stage 1 training. (**B**) Percentage freezing to the preconditioned sound (left panel) and to the conditioned light (right panel), averaged across the eight trials of their respective tests. Data shown are means ± SEM, and Cohen's d (d) for statistically significant results (contrasts used to compare Group VEH versus Group CHX). For raw data, see the *Figure 2—source data 1*.

DOI: https://doi.org/10.7554/eLife.47085.004

The following source data is available for figure 2:

**Source data 1.** Protein synthesis in the PRh is required for consolidation of a sensory sound-light memory.
DOI: https://doi.org/10.7554/eLife.47085.005

During conditioning and testing, the baseline levels of freezing were low (<10%) and did not significantly differ between groups (largest $F = 0.544$; $p = 0.471$). Acquisition of freezing to the light was successful. The mean (± SEM) levels of freezing for the final light trial were 85 ± 6.3% in Group VEH, and 76 ± 7.8% in Group CHX. Freezing to the light increased linearly across its pairings with shock ($F_{(1,16)} = 198.245$, $p < 0.0001$; 95% CI: [52.878, 71.622]). The rate of increase did not significantly differ between-groups (linear × group interaction: $F = 1.410$; $p = 0.301$), and there was no significant difference in their overall levels of freezing ($F = 1.102$; $p = 0.309$).

## Experiment 3: Fear of the preconditioned sound requires activity in the PRh during and after light-shock pairings

As noted above, there are two ways that sound-light and light-shock memories could be integrated so that the sound elicits freezing at test (*Figure 3*). The first is through mediated conditioning of the sound across pairings of the light and shock in stage 2 (online integration). The idea is that the light retrieves its sound associate from long-term memory and thereby allows the sound to associate with the shock. The rats thus form two associations in stage 2: a direct association between the presented light and shock, and an indirect or mediated association between the retrieved (but not presented) sound and shock. Integration, therefore, occurs online to imbue the innocuous sound with emotional value. The second way in which integration could occur is through chaining of the sound-light and light-shock memories at the time of testing. The idea here is that test presentations of the sound activate its now feared associate, the light, and thereby elicit freezing in anticipation of that feared

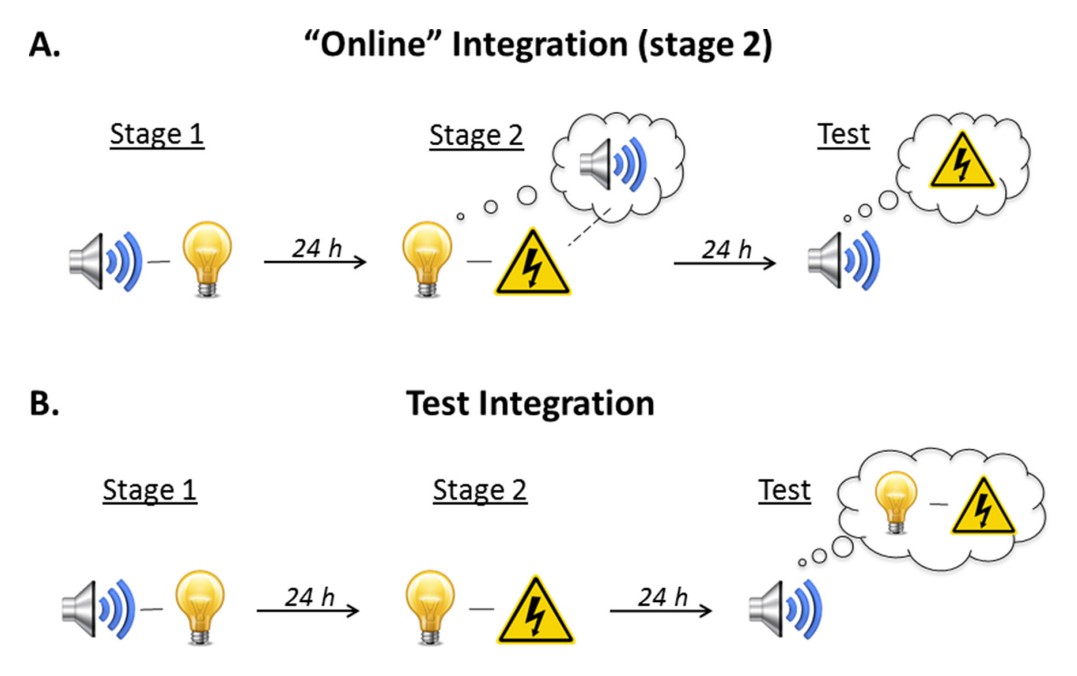

**Figure 3.** Illustration of the two possible mechanisms underlying integration of the sound-light memory formed in stage 1 and the light-shock memory formed in stage 2. (**A**) The first possibility is that the memories are integrated 'online' during training. When the subject is exposed to light-shock pairings in stage 2, presentation of the light triggers retrieval of its past associate, the sound, thereby allowing it to enter into an association with the shock. Test presentations of the sound then retrieve this mediated sound-shock association, and thus, elicit fear. (**B**) The second possibility is that the memories are integrated at the time of testing ('memory chaining'). According to this account, the sound-light memory (stage 1) and the light-shock memory (stage 2) are formed independently of each other. It is at the point of testing, when the sound is once again encountered, that the subject retrieves the initial sound-light memory and integrates (or chains) it with the light-shock memory; thus, the sound elicits fear.
DOI: https://doi.org/10.7554/eLife.47085.006

outcome. Integration, therefore, occurs when the sound-light and light-shock memories are required for action.

In order to distinguish between these alternatives, we trained rats in our standard protocol but silenced neural activity in the PRh, via infusions of the sodium channel blocker, bupivacaine, either before or after the light-shock pairings in stage 2 (*Figure 4A*). We anticipated that silencing the PRh would not affect formation of the light-shock fear memory, which is encoded and stored in the BLA. Hence, if freezing to the sound is based on a chaining of the sound-light and light-shock memories at the time of testing, this manipulation should spare that freezing as rats enter testing with both of those memories intact. If, however, freezing to the sound is based on a mediated sound-shock association formed across the light-shock pairings in stage 2, silencing neural activity in the PRh before (Group Pre-BUP) or after (Group Post-BUP) the light-shock pairings should undermine the ability of the light to retrieve its sound associate, and thereby, formation of the mediated sound-shock association. Hence, Groups Pre-BUP and Post-BUP should freeze less when tested with the sensory pre-conditioned sound than control rats (Group VEH).

The baseline levels of freezing during conditioning and test sessions were low (<10%), and did not significantly differ between the groups (largest $F$ = 0.915; p = 0.351). Fear conditioning of the light was successful. The mean (± SEM) levels of freezing to the light on its final pairing with shock were 90 ± 6.6% in Group VEH, 97 ± 2.9% in Group PRE-BUP, and 68 ± 15.6% in Group POST-BUP. There was a significant linear increase in freezing across the four light-shock pairings ($F_{(1,19)}$ = 106.682; p < 0.0001; 95% CI: [51.053, 77.002]). The groups did not differ in their overall levels of freezing (largest $F$ = 1.143; p = 0.298) and there were no linear trend × group interactions (largest $F$ = 4.252; p = 0.053).

*Figure 4B* shows the mean (± SEM) test level of freezing to the sound (left panel) and to the light (right panel; see also *Figure 4—source data 1*). It shows, as expected, that PRh infusions of

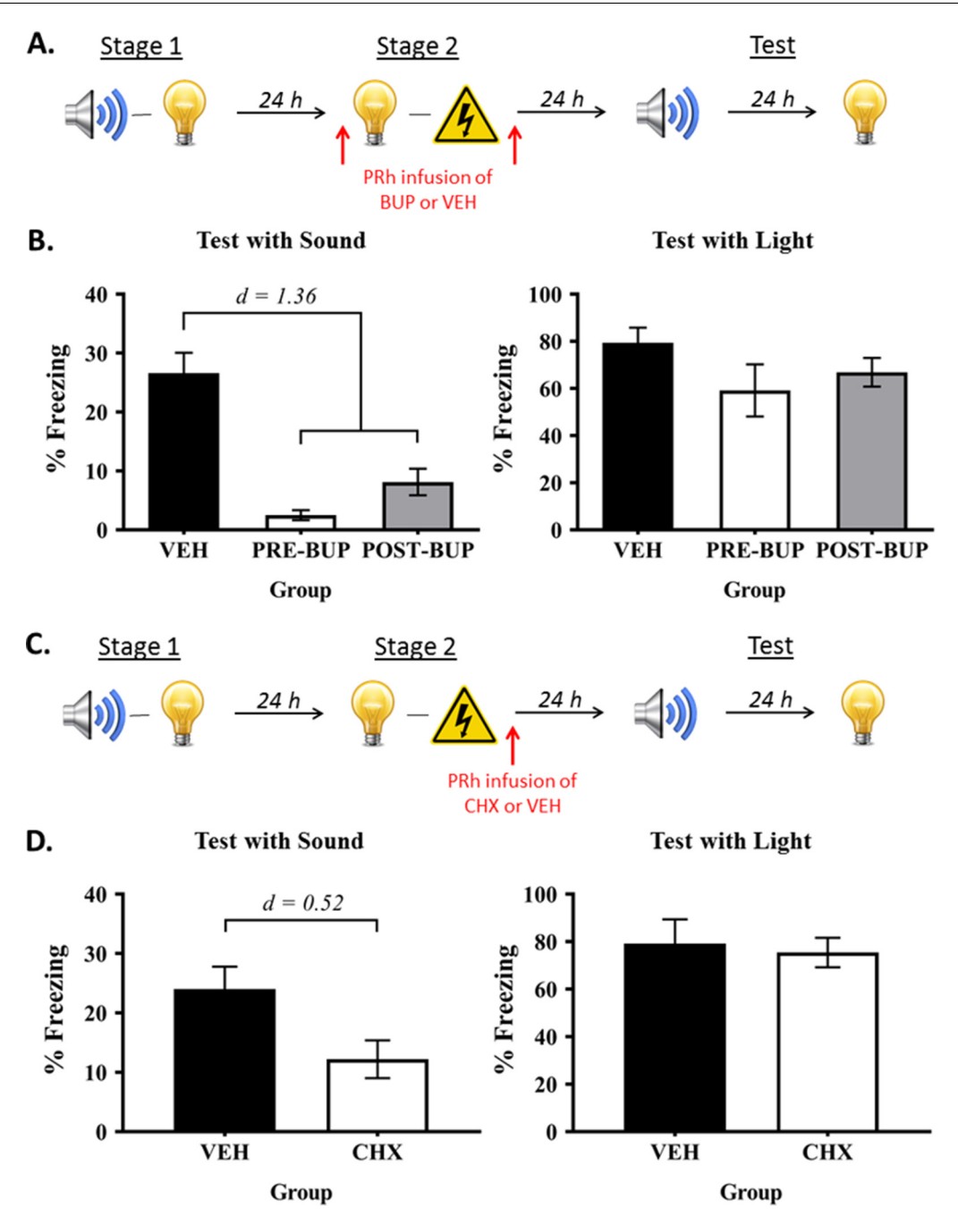

**Figure 4.** The PRh supports online integration of the sound-light and light-shock memories. (A) Behavioral procedure for Experiment 3 (Group VEH, $n = 8$; Group PRE-BUP, $n = 6$; and Group POST-BUP, $n = 8$). The red arrows indicate that the infusions of bupivacaine (BUP) or vehicle (VEH) occurred either before stage 2 or after stage 2. (B) Percentage freezing to the preconditioned sound (left panel) and to the conditioned light (right panel), averaged across the eight trials of their respective tests. Data shown are means ± SEM. The levels of freezing in both tests were analyzed using a set of planned orthogonal contrasts (*Hays, 1963*). The first contrast compared Group VEH versus the weighted average of Groups PRE-BUP and POST-BUP; and the second contrast compared Group PRE-BUP versus Group POST-BUP. Cohen's d (*d*) is shown for statistically significant results. (C) Schematic of the behavioral procedure for Experiment 4 (Group VEH, $n = 6$; and Group CHX, $n = 6$). The red arrow indicates the infusions occurred immediately after stage 2. (D) Percentage freezing to the preconditioned sound (left panel) and to the conditioned light (right panel), averaged across the eight trials during their respective tests. Data shown are means ± SEM, and Cohen's d (*d*) for statistically significant results. It is worth noting that the level of

*Figure 4 continued on next page*

*Figure 4 continued*

freezing to the sound in each of the treatment groups was not significantly different from the baseline (*F*s < 1). We take this result to mean that the infusion of bupivacaine or cycloheximide into the PRh before/after stage 2 training completely blocked formation of the mediated sound-shock association. For raw data, see the ***Figure 4— source data 1***.

DOI: https://doi.org/10.7554/eLife.47085.007

The following source data is available for figure 4:

**Source data 1.** The PRh supports online integration of the sound-light and light-shock memories.

DOI: https://doi.org/10.7554/eLife.47085.008

bupivacaine spared conditioning of the light, regardless of whether they occurred before (Group Pre-BUP) or after (Group Post-BUP) the light-shock pairings in stage 2: levels of freezing to the light in Group VEH did not significantly differ from Groups PRE- and POST-BUP (*F* = 3.174; p = 0.091), and there was no significant difference between the levels of freezing to the light in the latter two groups (*F* = 0.478; p = 0.498). However, consistent with an online integration mechanism, those infusions disrupted the test levels of freezing to the preconditioned sound: rats in Group VEH froze more to the sound than those in Groups PRE- and POST-BUP ($F_{(1,19)}$ = 44.64; p < 0.0001; *d* = 1.36; 95% CI: [14.593, 27.907]), who did not differ significantly from each other (*F* = 2.123; p = 0.161). These results confirm that the PRh is not involved in the encoding or consolidation of the light-shock fear memory, and show for the first time that the memory integration that supports freezing to the preconditioned sound occurs during formation of the light-shock memory in stage 2.

## Experiment 4: Consolidation of the mediated sound-shock association requires de novo protein synthesis in the PRh after light-shock conditioning

This experiment examined whether the effect of post-stage 2 PRh bupivacaine infusions could be reproduced when cycloheximide was infused into the PRh after the light-shock pairings in stage 2 (*Figure 4C*). Rats were exposed to sound-light pairings in stage 1, and to light-shock pairings in stage 2. Immediately after the stage 2 session, rats were infused with either cycloheximide (Group CHX) or vehicle alone (Group VEH) into the PRh. All rats were then tested with presentations of the sound and light in stage 3.

During fear conditioning and testing, the baseline levels of freezing were low (<10%) and did not differ between the two groups (largest *F* = 1.888; p = 0.199). Fear conditioning of the light was successful. There was a significant linear increase in freezing across each light-shock pairings ($F_{(1,10)}$ = 56.343; p < 0.0001; 95% CI: [40.432, 74.568]). The groups did not differ in their rate of acquisition (linear x group interaction, *F* = 0.757; p = 0.405), or in their overall freezing to the light (*F* = 0.458; p = 0.514).

The mean (± SEM) test levels of freezing to the sensory preconditioned sound and to the conditioned light are shown in the left and right panels, respectively, of *Figure 4D*. Relative to vehicle-infused controls, rats that received the PRh cycloheximide infusions after the light-shock pairings froze just as much during test presentations of the light (*F* = 0.098; p = 0.761), but less during test presentations of the preconditioned sound ($F_{(1,10)}$ = 5.690; p = 0.038; *d* = 0.52; 95% CI: [0.778, 22.833]). These results again confirm that the PRh is not involved in the encoding or consolidation of the light-shock fear memory, and show that online integration of sound-light and light-shock memories requires de novo protein synthesis in the PRh.

## Experiment 5: Activity in the PRh, but not protein synthesis, is required for the expression of fear to the sensory preconditioned sound

The final experiment examined whether the PRh also plays a dissociable role in retrieval and/or expression of the directly conditioned light-shock memory versus the mediated conditioned sound-shock memory. Rats were first exposed to sound-light pairings in stage 1 and light-shock pairings in stage 2. They then received a PRh infusion of either bupivacaine, cycloheximide or vehicle immediately prior to test presentations of the preconditioned sound and of the directly conditioned light (*Figure 5A*).

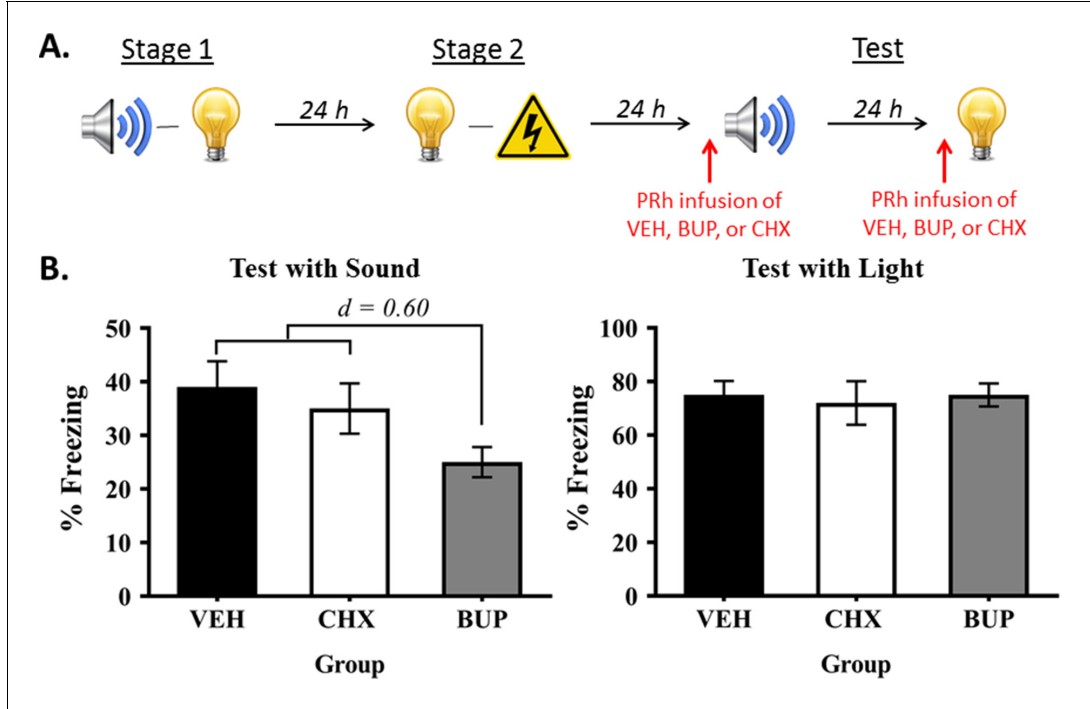

**Figure 5.** Neuronal activity in the PRh is required for expression of fear to the preconditioned sound, but not the conditioned light. (**A**) Schematic of the procedure for Experiment 5 (Group VEH, *n* = 10; Group CHX, *n* = 7; and Group BUP, *n* = 7). The red arrows indicate that infusions occurred before the test for the sensory preconditioned sound and before the test for the conditioned light. (**B**) Percentage freezing to the preconditioned sound (left panel) and to the conditioned light (right panel), averaged across the eight trials of their respective tests. Data shown are means ± SEM. The levels of freezing in both tests were analyzed using a set of planned orthogonal contrasts (*Hays, 1963*). The first contrast compared Group BUP versus the weighted average of Groups VEH and CHX (both of which were expected to show fear); and the second contrast compared Group VEH versus Group CHX. Cohen's d (*d*) is shown for statistically significant results. For raw data, see the *Figure 5—source data 1*.
DOI: https://doi.org/10.7554/eLife.47085.009

The following source data is available for figure 5:

**Source data 1.** Neuronal activity in the PRh is required for expression of fear to the preconditioned sound, but not the conditioned light.
DOI: https://doi.org/10.7554/eLife.47085.010

The baseline levels of freezing during the fear conditioning and testing sessions were low (<10%) and did not significantly differ between the groups (largest *F* = 1.279; p = 0.271). Fear conditioning of the light was successful. The mean (± SEM) levels of freezing to the light on its final pairing with shock were 84 ± 8.3% in Group VEH, 80 ± 12% in Group BUP, and 77 ± 13% in Group CHX. All rats froze across the light-shock pairings, and there were no between-group differences in their overall levels of freezing to the light (largest *F* = 2.107; p = 0.161) or in their rates of acquisition (largest *F* = 0.017; p = 0.898).

The test levels of freezing (mean ± SEM) to the sound and to the light are shown in the left and right panels, respectively, of *Figure 5B* (see also *Figure 5—source data 1*). These tests confirmed that neuronal activity in the PRh is not required for retrieval of the light-shock memory and/or its expression in freezing, as rats in the three groups exhibited similar levels of freezing when tested with the light (largest *F* = 0.140; p = 0.712). The tests also showed that activity in the PRh is required for freezing to the preconditioned sound, as rats that received a pre-test infusion of bupivacaine into the PRh froze significantly less to the sound than rats infused with vehicle or cycloheximide ($F_{(1,21)}$ = 4.552; p = 0.045; *d* = 0.60; 95% CI: [−24.037,–0.308]), who did not differ from each other in their levels of freezing (*F* = 0.296; p = 0.592). Hence, the PRh is selectively required for retrieval and/or expression of freezing to the preconditioned sound, and this retrieval/expression occurs independently of de novo protein synthesis in the PRh.

## Discussion

The experiments reported here provide further evidence that sound-light and light-shock memories have distinct substrates within the MTL. Previous work had shown that formation of the sound-light memory requires activation of NMDAr in the PRh (*Holmes et al., 2013*), and that the consolidation of this memory requires ERK/MAPK signaling in the PRh (*Holmes et al., 2018*). The present work shows that this consolidation also requires de novo protein synthesis in the PRh (Experiment 2). In contrast, the PRh is not required for encoding, consolidating or retrieving information about a light-shock fear memory (*Phillips and LeDoux, 1995*; *Romanski and LeDoux, 1992a*; *Romanski and LeDoux, 1992b*; *Wilensky et al., 2006*). Instead, the BLA is critical for each of these processes (*Johansen et al., 2011*): encoding of a light-shock fear memory requires activation of NMDAr in the BLA (*Campeau et al., 1992*; *Fanselow and Kim, 1994*; *Miserendino et al., 1990*); consolidation of this memory requires neuronal activity in this region, including de novo protein synthesis (*Maren et al., 2003*; *Schafe and LeDoux, 2000*); and retrieval/expression of a light-shock fear memory requires neuronal activity in the BLA, but occurs independently of de novo protein synthesis (*Abel and Lattal, 2001*; *Bourtchouladze et al., 1998*; *Helmstetter and Bellgowan, 1994*; but see *Lopez et al., 2015*).

Given the distinct substrates of the sound-light and light-shock memories in the PRh and BLA, respectively, the major question of interest asked here was *when* these memories are integrated so that the sound, which is never paired with shock, comes to elicit the defensive or fear response of freezing. To address this question, we examined whether manipulations of the PRh during formation of the light-shock fear memory influenced the test level of freezing to the preconditioned sound. The results showed that pharmacological inhibition of activity in the PRh before light-shock pairings spared freezing to the directly conditioned light, but reduced freezing to the preconditioned sound. They additionally showed that inhibition of PRh activity after light-shock pairings, including inhibition of de novo protein synthesis, produced the same pattern of test results. These manipulations should have left intact the PRh dependent sound-light memory formed in stage 1 and the BLA dependent light-shock memory formed in stage 2. Hence, if these memories were integrated at the time of testing (via the memory chaining mechanism described above), then independently of our PRh manipulations, the sound should have activated its now dangerous associate, the light, and elicited freezing. The fact that this did not occur indicates that integration had occurred in advance of testing, when the light was rendered dangerous via its pairing with shock (see also *Coureaud et al., 2013*). Nonetheless, we acknowledge that even these results do not completely rule out the possibility that some portion of the integration occurs at the time of testing in stage 3. For example, given the initial coding of the light in the PRh in stage 1 (as part of the sound-light association), it is possible that the light-shock association that forms in stage two is coded in parallel in the amygdala (principally) and PRh. If so, the PRh may further support responding to the sensory preconditioned sound by integrating the sound-light and light-shock associations at the point of testing (or at least the portions of these associations that are represented in the PRh). According to this account, silencing the PRh during stage 2 would have prevented coding of the light-shock association in this region, thereby undermining the possibility of memory chaining during testing in stage 3; and silencing the PRh during stage 3 would have blocked memory chaining directly, thereby disrupting responding to the preconditioned sound.

The present study used a between-subject design to show that freezing to the sensory preconditioned sound is doubly contingent on sound-light and light-shock pairings across the two stages of training; and that the PRh codes a mediated sound-shock association during stage 2 conditioning of the light. It does, however, leave open the question of whether the PRh codes for other types of mediated associations. This question could be assessed using a within-subject design, in which rats are exposed to two sets of stimulus pairings, for example tone-light and noise-flashing light (flash), in stage 1; shocked presentations of the light and non-shocked presentations of the flash in stage 2; and test presentations of the tone and noise in stage 3. Sensory preconditioning would be evident as greater test responding to the tone than the noise (e.g., *Jones et al., 2012*); see also *Sharpe et al., 2017*). Such a difference could originate in two distinct associations formed across stage 2 of training: an excitatory light-shock association and an inhibitory flash-no shock association. That is, just as the shocked presentations of the light allow the tone to enter into an association with the shock, non-reinforced presentations of the flash should allow the noise to enter into an

inhibitory, no shock association. In other words, the test differences between the tone and the noise could be due to the former enhancing and the latter reducing freezing. Since both the tone-light and noise-flash associations are likely to be encoded in the PRh in stage 1, we predict that silencing activity in the PRh across the discriminative conditioning of the light and flash in stage 2 would eliminate the test differences between the tone and the noise. It would do so by preventing not only the mediated excitatory conditioning of the tone but also the mediated inhibitory conditioning of the noise.

It is worth noting that the within-subject design just described requires many trials to achieve differential responding in stage 2; that is, for the rats to freeze when presented with the shocked light and not to freeze when presented with the non-shocked flash. In contrast, relatively few shocked presentations were used here. This difference may be important with respect to when the information acquired in sensory preconditioning and conditioning is integrated. There is evidence that the likelihood of mediated conditioning decreases with the number of conditioning trials (*Holland, 1998*). Moreover, *Jones et al. (2012)* used just such a within-subject, sensory preconditioning design in an appetitive conditioning protocol (one that involved multiple trials to produce the discrimination in stage 2) to show that integration occurred at the time of testing: specifically, they reported that silencing the orbitofrontal cortex prior to testing reduced the expression of sensory preconditioned appetitive responding (see also *Sharpe et al., 2017*). Nonetheless, it remains to be shown within a single study that variations in the number of stage 2 conditioning trials regulates whether integration occurs in stage 2 (online) or stage 3 (memory chaining).

The present findings have also identified both commonalities and differences between sensory preconditioning and a conceptually-related protocol, second-order conditioning, in which animals are exposed to the same training but in the reverse order, that is light-shock pairings in stage 1 and sound-light pairings in stage 2. Like in sensory preconditioning, rats integrate the memories formed in the two stages of second-order conditioning so that the sound, which is not paired with shock, elicits freezing; and this integration occurs during training rather than at test (*Gewirtz and Davis, 2000*; *Parkes and Westbrook, 2010*; *Rizley and Rescorla, 1972*). However, the integration in second-order conditioning differs from that in sensory preconditioning in two critical respects. First, unlike in sensory preconditioning where the two component memories are encoded and stored in distinct regions of the MTL (*Holmes et al., 2013*), both of the component memories in second-order conditioning are encoded and stored in the BLA (*Holmes et al., 2013*; *Parkes and Westbrook, 2010*). Second, unlike in sensory preconditioning, where consolidation of the integrated (or mediated) sound-shock memory requires de novo protein synthesis in the PRh, consolidation of the integrated second-order memory occurs independently of de novo protein synthesis in the BLA (*Lay et al., 2018*; *Leidl et al., 2018*). These differences show that the order in which information is presented (or processed) determines how distinct memories are encoded and stored in the MTL, as well as the mechanisms involved in their integration. The integration of memories stored in distinct MTL regions (the PRh and BLA in sensory preconditioning) requires cellular changes that are supported by de novo protein synthesis; whereas the integration of memories stored within the same MTL region (the BLA in second-order conditioning) does not require this synthesis (*Lay et al., 2018*; *Leidl et al., 2018*).

The present findings also extend previous work showing that contextual information encoded and stored in one region of the brain can be retrieved and integrated with motivational information processed elsewhere (*Wolosin et al., 2012*; *Zeithamova et al., 2012*): for example memories of specific contexts are encoded and stored in distinct regions of the hippocampus, and when activated (through stimulation of context-specific ensembles, or exposure to reminder cues), can associate with nociceptive information in the amygdala to form a context fear memory (*Bae et al., 2015*; *Matus-Amat et al., 2007*; *Ramirez et al., 2013*; *Rudy and O'Reilly, 2001*). They extend these findings by showing that animals can integrate a distinct sensory element of one memory (not just context elements) with motivational components of another (even when the two have never been experienced together); and that this integration involves protein synthesis-dependent changes in the PRh. This is generally consistent with current theories of MTL function (for reviews, see *Squire et al., 2004*; and *McGaugh et al., 2002*) which hold that: (i) context and stimulus elements are represented in a distributed network of regions, (ii) the values of these representations are updated across the course of experience (i.e., 'on the fly' rather than at choice points), and (iii) the amygdala plays a critical role in this updating process. However, the results of the final experiment suggest that these

theories require further elaboration: retrieval/expression of the mediated sound-shock memory required activity in the PRh, whereas retrieval/expression of the directly conditioned light-shock memory did not. This difference implies that, as a consequence of its direct pairings with shock, the light was recoded so that responding to this stimulus ceased to depend on the PRh. It is consistent with previous work which showed that danger shifts the encoding and consolidation of innocuous sensory information from the PRh to the BLA (*Holmes et al., 2018*; *Holmes and Westbrook, 2017*). Thus, in addition to its role as a modulator of PRh-dependent stimulus representations, the BLA itself encodes and recodes stimulus representations when danger is present; and thereafter, remains involved in the retrieval/expression of information about those stimuli.

In summary, using a sensory preconditioning protocol, the present study provides evidence that the brain integrates sound-light and light-shock memories across the course of training; and that the cortical region that codes for the integration, the PRh, remains critical for the subsequent retrieval and expression of the integrated memory. As noted above, our findings do not preclude the possibility that some portion of the integration in the present study occurred at the point of testing; or indeed, that there are circumstances under which sensory and emotional memories are exclusively integrated at the time of testing (*Rizley and Rescorla, 1972*; *Sadacca et al., 2018*; *Sharpe et al., 2017*). They do, however, suggest that with very simple procedures of the sort used here, the bulk of the integration occurs during training. More generally, our findings reiterate that memory is not a repository of experience, but rather, a process of construction and reconstruction. These constructions are not a limitation of the system, but rather, a feature of its design: they help animals and people to solve novel problems in the absence of further experience, and thereby, adapt to changes in the environment.

## Materials and methods

### Subjects

Subjects were experimentally naïve male and female Long-Evan rats, obtained from a colony maintained by the Biological Resources Centre at the University of New South Wales, and Sprague-Dawley rats, obtained from a commercial supplier (Animal Resources Centre, Perth, Australia). The rats were housed in plastic cages (40 cm width × 22 cm height x 67 cm length), with a minimum of four rats per cage. These cages were kept in a colony room where the temperature was maintained at approximately 21°C, and kept on a 12 hr light/dark cycle (lights on at seven am and off at seven pm). All rats received unrestricted access to food and water for the duration of the experiment.

### Surgery and drug infusions

Prior to behavioral training and testing, rats were surgically implanted with bilateral cannulas targeting the perirhinal cortex (PRh). Rats received an intraperitoneal (i.p.) injection of a combination of 1.3 ml/kg of the anesthetic, ketamine (Ketapex; Apex Laboratory), at a concentration of 100 mg/ml, and 0.3 ml/kg of the muscle relaxant, xylazine (Rompun; Bayer), at a concentration of 20 mg/ml. The rat was then mounted onto a stereotaxic apparatus (David Kopf Instruments) and incisions made over the skull. Two holes were drilled through the skull and 26-gauge guide cannulas (Plastic Ones) were implanted into the brain, one in each hemisphere. The tips of the cannulas targeted the PRh at coordinates 4.15 mm posterior to Bregma, 5.00 mm lateral to the midline, 8.4 mm ventral to Bregma, and angled at approximately 9° (*Paxinos and Watson, 2006*). Guide cannulas were secured in place with four jeweller's screws and dental cement. A dummy cannula was kept in each guide cannula at all times except during drug infusions. Immediately after surgery, rats received an i.p. injection with a prophylactic (0.4 ml) dose of a 300 mg/kg solution of procaine penicillin. Rats were allowed 7 d to recover from surgery, during which they were monitored and weighed daily.

Bupivacaine, cycloheximide or vehicle was infused bilaterally into the PRh. For these infusions, infusion cannulas were connected to 25 µl Hamilton syringes via polyethylene tubing. These syringes were fixed to an infusion pump (Harvard Apparatus). The infusion procedure began by removing the dummy caps from the guide cannulas on each rat and inserting 33-gauge infusion cannulas in its place. The pump was programmed to infuse a total of 0.5 µl at a rate of 0.25 µl/min, which resulted in a total infusion time of 2 min. The infusion cannulas remained in place for an additional 2 min after the infusion was complete to allow for diffusion of the drug into the PRh tissue and to avoid

reuptake of the drug. This resulted in a total infusion time of 4 min. After the additional 2 min, the infusion cannulas were removed and replaced with dummy cannulas. The day prior to infusions, the dummy cannula was removed and the infusion pump was activated to familiarize the rats with the procedure and thereby minimize any effect of procedure on the day of infusions.

## Drugs

The sodium channel blocker, bupivacaine hydrochloride (0.5% w/v), was obtained from Cenvet Australia. Nonpyrogenic saline (0.9% w/v) was used as a vehicle solution. The protein synthesis inhibitor, cycloheximide, was obtained from Sigma Australia and prepared in the manner described by *Duvarci et al. (2005)*. Briefly, it was dissolved in 70% ethanol to create a stock solution of 200 µg/µl. The stock was then diluted 1:4 in ACSF to yield a final solution of 40 µg/µl. A 70% ethanol solution that was diluted 1:4 in ACSF was used as the vehicle solution. Previous studies have shown that the level of protein synthesis following an amygdala infusion of cycloheximide (similar dose to the current study) returns to baseline levels after 24 hr (*Kesner et al., 1981*; see also *Kleim et al., 2003*); and that the effects of intracerebral bupivacaine infusions are short-lasting, ranging from minutes to a maximum of 2 hr (see *Caterall and Mackie, 1996*; *Haralambous and Westbrook, 1999*).

## Histology

After behavioral training and testing, rats were euthanized with a lethal dose of sodium pentobarbital. The brains were extracted and sectioned into 40 µl coronal slices. Every second slice was mounted onto a glass microscope slide and stained with cresyl violet. The placement of the cannula tip was determined under a microscope using the boundaries defined by *Paxinos and Watson (2006)*. Rats with misplaced cannulas were excluded from statistical analysis. *Figure 6* shows placement of the most ventral portion of these cannulas in the PRh for all rats that were included in the statistical analyses of data from Experiments 2—5. The numbers of rats excluded from each experiment based on misplaced cannulas were five rats in Experiment 2, seven rats in Experiment 3, two rats in Experiment 4, and four rats in Experiment 5. The final $n$s for each group are presented in the figure legend for each experiment.

## Behavioral apparatus

All experiments were conducted in four identical chambers (30 cm width $\times$ 26 cm length $\times$ 30 cm height). The side walls and ceiling were made of aluminum, and the front and back walls were made of clear plastic. The floor consisted of stainless steel rods, each 2 mm in diameter and spaced 13 mm apart (center-to-center). A waste tray containing bedding material was located under the chamber floor. At the end of each session, the chambers were cleaned with water and any soiled bedding was removed from the waste trays and replaced with fresh bedding.

Each chamber was enclosed in a sound- and light-attenuating wooden cabinet. The cabinet walls were painted black. A speaker and LED lights within a fluorescent tube were mounted onto the back wall of each cabinet. The speaker was used to deliver a 1000 Hz square-wave tone stimulus, presented at 70–75 dB when measured at the center of the chamber (digital sound meter: Dick Smith Electronics, Australia). The LED lights were used to deliver a flashing light stimulus, presented at 3.5 Hz. A custom-built, constant current generator was used to deliver a 0.5 s duration, 0.8 mA intensity shock to the floor of each chamber.

Each chamber was illuminated with an infrared light source, and a camera mounted on the back wall of each cabinet was used to record the behavior of each rat. The cameras were connected to a monitor and DVD recorder located in an adjacent room. This room also contained the computer that controlled stimulus and shock presentations through an interface and appropriate software (MatLab, MathWorks).

## Behavioral procedure

### Context exposure

On Days 1 and 2, rats received two 20 min exposures to the chambers, one in the morning and the other approximately 3 hr later in the afternoon.

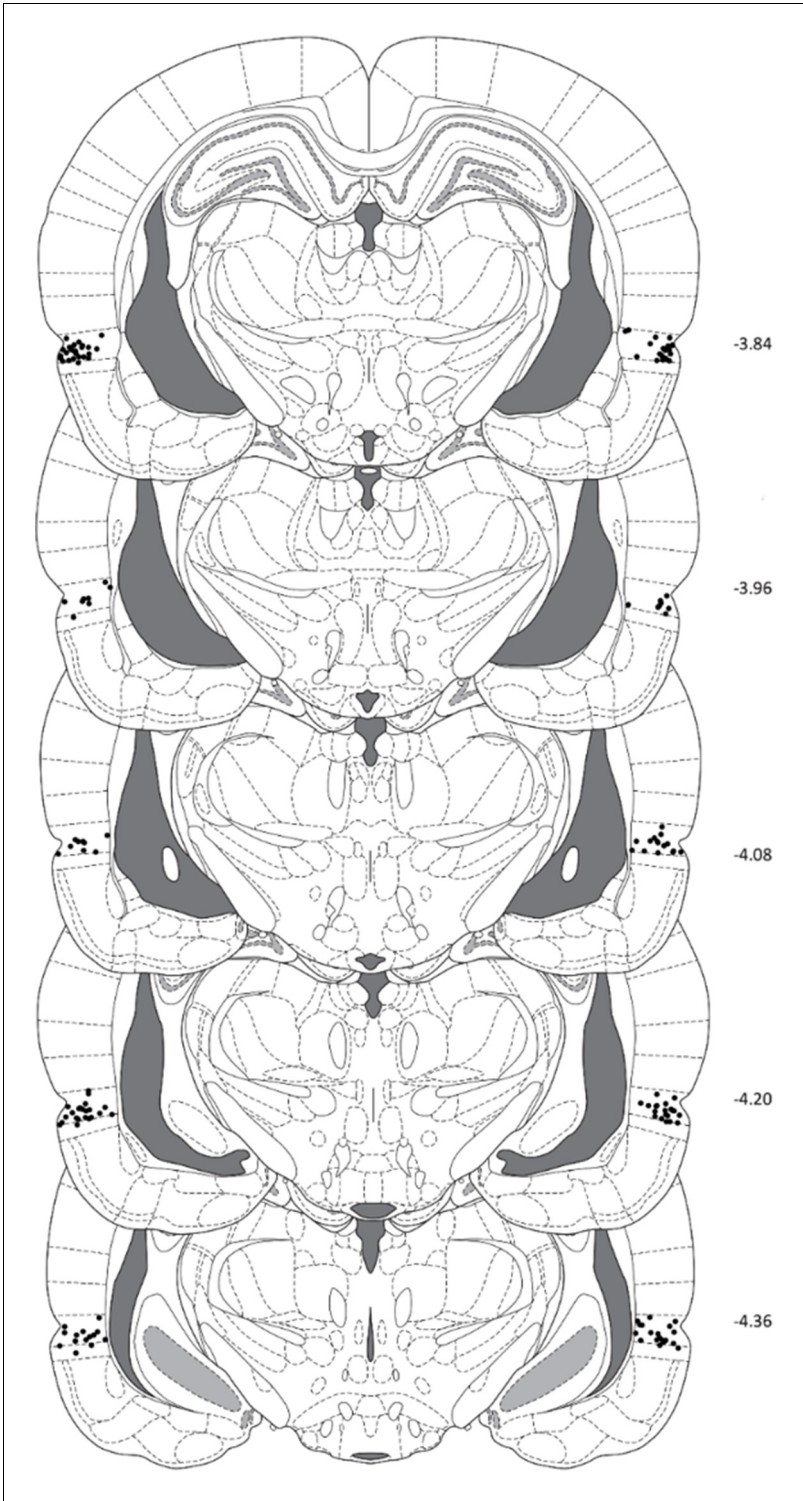

**Figure 6.** Cannula placements in the PRh taken from rats in Experiments 2–5. The most ventral portion of the cannulas are marked on coronal sections based on the atlas of *Paxinos and Watson (2006)*.
DOI: https://doi.org/10.7554/eLife.47085.011

## Sensory preconditioning

On Day 3, all rats received eight presentations of the sound and eight of the light. Each presentation of the sound was 30 s in duration and each presentation of the light was 10 s in duration. The offset of one stimulus co-occurred with the onset of the other stimulus for groups that received paired presentations of the sound and the light, while these stimuli were presented separately for groups that received explicitly unpaired presentations. The first stimulus presentation occurred 5 min after rats were placed into the chambers, and the interval between each paired presentation was fixed at 5 min while the interval between each separately presented stimulus was fixed at 150 s. After the last stimulus presentation, rats remained in the chambers for an additional 1 min. They were then returned to their home cages.

It should be noted that the sound and the light were fully counterbalanced for all experiments. That is, for half of the rats in each group, the sound would be the preconditioned stimulus, while the light would be the conditioned stimulus; and for the remainder, the light would be the preconditioned stimulus, and the sound would be the conditioned stimulus. However, for convenience of explanation, the following designs will be described with reference to one half of the counterbalancing.

## First-order conditioning

On Day 4, all rats received four presentations of the light and four presentations of footshock. The first light presentation occurred 5 min after rats were placed in chambers. In groups that received paired presentations, each 10 s light co-terminated with footshock, and the interval between light-shock pairings was 5 min. In groups that received explicitly unpaired presentations, presentations of the light and the footshock occurred separately, 165 s apart. Rats remained in the chambers for an additional 1 min after the final stimulus presentation and were then returned to their home cages.

## Context extinction

On Day 5, rats received two 20 min exposures to the chambers, one in the morning and the other in the afternoon. These exposures were intended to extinguish any freezing elicited by the context and thereby provide a measure of the level of freezing to the sound and the light un-confounded by context-elicited freezing. On Day 6, rats received a further 10 min extinction exposure to the context.

## Test

On Day 6, approximately 2 hr after the context-extinction session, rats were tested for their levels of freezing to the preconditioned sound. On Day 7, rats were tested for their levels of freezing to the conditioned light. For each test, the first stimulus was presented 2 min after rats were placed in the chambers. Each test consisted of eight stimulus alone presentations, with a 3 min interval between each presentation. Each sound presentation was 30 s in duration and each light was 10 s. Rats remained in the chamber for an additional min after the final stimulus presentation in each test.

## Scoring and statistics

Conditioning and testing sessions were recorded on DVD. Freezing, defined as the absence of all movements except those required for breathing (*Fanselow, 1980*), was used as a measure of conditioned fear. Freezing data were collected using a time-sampling procedure in which each rat was scored as either 'freezing' or 'not freezing' every 2 s by an observer blind to the rat's group allocation. A second blind observer scored a randomly selected 25% of the data. The correlation between the scores were high (Pearson > 0.9). The data were analyzed using a set of planned orthogonal contrasts (*Hays, 1963*), with the type one error rate controlled at $\alpha = 0.05$. Standardized 95% confidence intervals (CIs) were reported for significant results, and Cohen's d (*d*) is reported as a measure of effect size (where 0.2, 0.5, and 0.8 is a small, medium and large effect size respectively). The required number of rats per group was determined during the design stage of the study. It was based on our prior experience in running sensory preconditioning studies: experiments of this sort typically require an average of 8 subjects per group (or per comparison in the contrast testing procedure) to provide sufficient statistical power to detect effect sizes > 0.5 with the recommended probability of 0.8–0.9.

## Acknowledgements

The authors thank Dr. Kelly Clemens and Dr. Vincent Laurent for helpful advice in development of the experimental program.

## Additional information

### Funding

| Funder | Grant reference number | Author |
|---|---|---|
| Australian Research Council | DP170103952 | Nathan M Holmes |
| National Health and Medical Research Council | APP1146999 | Nathan M Holmes |

The funders had no role in study design, data collection and interpretation, or the decision to submit the work for publication.

### Author contributions

Francesca S Wong, Data curation, Formal analysis, Writing—original draft; R Fred Westbrook, Conceptualization, Supervision, Funding acquisition, Writing—review and editing; Nathan M Holmes, Conceptualization, Resources, Supervision, Funding acquisition, Writing—original draft, Writing—review and editing

### Author ORCIDs

Francesca S Wong (iD) https://orcid.org/0000-0002-8533-9833
Nathan M Holmes (iD) https://orcid.org/0000-0002-0592-2026

### Ethics

Animal experimentation: This study was performed in strict accordance with the guidelines published by the National Health and Medical Research Council in Australia. All of the animals were handled according to approved Animal Care and Ethics Committee (ACEC) protocols at the University of New South Wales (Permit Number: ACEC 17/139A). All surgery was performed under ketamine-xylazine induced anesthesia, and every effort was made to minimize suffering.

### Decision letter and Author response

Decision letter https://doi.org/10.7554/eLife.47085.014
Author response https://doi.org/10.7554/eLife.47085.015

## Additional files

### Supplementary files

• Transparent reporting form
DOI: https://doi.org/10.7554/eLife.47085.012

### Data availability

All data generated or analysed during this study are included in the manuscript and supporting files. Source data files have been provided for Figures 1, 2, 4 and 5.

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
