## [Decision Letter]

Thank you for submitting your article "Online integration of sensory and emotional (fear) memories in the medial temporal lobe" for consideration by *eLife*. Your article has been reviewed by three peer reviewers, and the evaluation has been overseen by a Reviewing Editor and Timothy Behrens as the Senior Editor. The following individuals involved in review of your submission have agreed to reveal their identity: Matthew PH Gardner (Reviewer #2).

The reviewers have discussed the reviews with one another and the Reviewing Editor has drafted this decision to help you prepare a revised submission.

Summary:

This manuscript addresses an important question of how distinct memories are integrated in order to guide future behavior. The authors use the sensory preconditioning paradigm in which a neutral stimulus acquires motivational significance through indirect association with an emotionally salient stimulus (shock). They find that perirhinal cortex must be online and capable of protein synthesis during the cue-shock learning. This learning allows for the expression of the inferred fear to the non-shock paired neutral stimulus at the time of test, and critically, disrupting this learning has no effect on the direct cue-shock learning. This suggests that sensory preconditioning relies on mediated learning during cue-shock learning, and that perirhinal cortex is necessary for this indirect learning between the unpaired neutral stimulus and the emotionally relevant stimulus. Furthermore, the authors show that activation of this inferred association is required at the time of test. This suggests that perirhinal cortex is necessary for both the learning of the indirect association and the expression of this learning during integration.

All reviewers agreed that this is a well-written manuscript implementing a straightforward and thorough experimental design with clear results. However, all reviewers also agreed that the current results do not rule out a chaining account and that alternative explanations for the findings should be discussed.

Essential revisions:

1) As the authors mention in the Introduction, there is considerable evidence that integration may occur at the time of test (e.g., Jones et al., 2012). Can the authors elaborate on what experimental conditions might lead to either mediated learning or integration at test? It seems unlikely that this would simply be due to use of either a positive or negative reinforcer used in the design. If this is true, what are the conditions that might promote one method over the other?

2) Assuming the systems that allow for mediated learning vs. chaining are independent, it is interesting that they are not redundant. That is, why does inactivation of perirhinal cortex at the time of test disrupt behavior so effectively? If the systems are active in parallel, then the chaining system should be intact and allow for integration even when the mediated system is inactive. In other words, it is not clear why perirhinal activity would be necessary if the fear was integrated with the sensory association during stage 2. This somewhat undercuts the main conclusion because this finding is more consistent with the chaining model dismissed by the authors. Specifically, the current results are compatible with the interpretation that perirhinal cortex is in fact part of the chaining system and therefore must have access to cue-cue and cue-shock associations during the test. If perirhinal cortex is offline in stage 1, it cannot consolidate cue-cue associations. If this region is inactive during stage 2, it cannot form cue-shock associations. And if it is inactive during test, it cannot perform the integration. Of note, other areas (e.g., BLA) may in parallel acquire the cue-shock association during stage 2 to support responding to the CS at test, but this association cannot be used for chaining. The authors should consider this alternative interpretation of their data which supports a chaining mechanism rather than mediated learning.

3) To demonstrate the specificity of the effects, an important control condition would have been a second stimulus pair C-D, where D is not paired with shocks (in contrast to A-B). It would have been useful to see that the animals can discriminate between a shocked (B) and an unshocked (D) stimulus and only freeze in response to B. Likewise, in such a situation one would expect freezing in response to A, but not C. In addition, it would have been useful to see that the response to C is not affected by the various manipulations. While the reviewers agree that an additional control experiment is not strictly necessary, these issues should at least be discussed. For instance, is it possible that the absence of a C-D pair in the current design could explain the apparent dominance of mediated learning?

[Editors' note: further revisions were requested prior to acceptance, as described below.]

Thank you for resubmitting your work entitled "Online integration of sensory and emotional (fear) memories in the medial temporal lobe" for further consideration at *eLife*. Your revised article has been favorably evaluated by Timothy Behrens as the Senior Editor, and a Reviewing Editor, in consultation with the original reviewers.

The authors have adequately addressed essential points 1 and 3, in addition to all minor points. As a result, the manuscript has improved but there are some remaining issues that need to be addressed before acceptance. Specifically, the reviewers felt that the authors did not sufficiently respond to essential point 2, in which we asked for discussion of an alternative interpretation of the current findings.

During the initial round of reviews, all reviewers agreed that the current results do not rule out the idea that integration happens during phase 3. They felt that the current findings are also compatible with the idea that integration occurs in phase 3 (i.e., chaining). To recapitulate this point: It is possible that PRh encodes cue-cue associations during phase 1 and cue-outcome associations during phase 2, but that PRh is also critical for integrating cue-cue and cue-outcome associations during phase 3, while another area (e.g. amygdala) additionally stores cue-outcome associations during phase 2. If PRh is inactive during any of these three phases, responding to the indirectly associated cues will be disrupted in phase 3, even if integration happens exclusively in phase 3 (i.e., even if there is no mediated learning at all during phase 2). Specifically, inactivation during phase 2 will prevent encoding of cue-outcome associations which are required for integration in phase 3. In contrast, responding to directly experienced cue-outcomes associations in phase 3 is not impaired by inactivating PRh at any stage because these associations are additionally stored in the amygdala, but they cannot be used for integration. This idea would also explain why PRh inactivation during phase 3 disrupts responding to indirectly associated cues.

The authors did not mention this alternative interpretation in the new discussion paragraph. We would like to ask you to discuss this alternative interpretation of your results and to clearly state that the current results do not rule out the idea that integration happens during phase 3.

---

## [Author Response]

Essential revisions:1) As the authors mention in the Introduction, there is considerable evidence that integration may occur at the time of test (e.g., Jones et al. 2012). Can the authors elaborate on what experimental conditions might lead to either mediated learning or integration at test? It seems unlikely that this would simply be due to use of either a positive or negative reinforcer used in the design. If this is true, what are the conditions that might promote one method over the other?

We have included two new paragraphs in the Discussion that address the evidence for test integration, as well as experimental conditions that might shift the timing of integration from stage 2 (online) to stage 3 (testing). As these two paragraphs address each of the major points raised in the review, they have been copied and pasted below.

2) Assuming the systems that allow for mediated learning vs. chaining are independent, it is interesting that they are not redundant. That is, why does inactivation of perirhinal cortex at the time of test disrupt behavior so effectively? If the systems are active in parallel, then the chaining system should be intact and allow for integration even when the mediated system is inactive. In other words, it is not clear why perirhinal activity would be necessary if the fear was integrated with the sensory association during stage 2. This somewhat undercuts the main conclusion because this finding is more consistent with the chaining model dismissed by the authors. Specifically, the current results are compatible with the interpretation that perirhinal cortex is in fact part of the chaining system and therefore must have access to cue-cue and cue-shock associations during the test. If perirhinal cortex is offline in stage 1, it cannot consolidate cue-cue associations. If this region is inactive during stage 2, it cannot form cue-shock associations. And if it is inactive during test, it cannot perform the integration. Of note, other areas (e.g., BLA) may in parallel acquire the cue-shock association during stage 2 to support responding to the CS at test, but this association cannot be used for chaining. The authors should consider this alternative interpretation of their data which supports a chaining mechanism rather than mediated learning.

We agree that, under some conditions, integration occurs at the time of testing. We only show that, in our conditions, integration occurs during stage 2 of training. Rather than the two mechanisms of integration operating in parallel (which would not explain the results of our study), the most parsimonious explanation of our data is that the evidence for sensory preconditioned responding at test is *entirely* due to formation of a mediated association in stage 2; and that the requirement for PRh activity at the time of testing reflects its role in retrieval of the mediated association. We address this point, as well as studies showing that integration occurs at the time of testing, in the third and fourth paragraphs of the Discussion (see below).

3) To demonstrate the specificity of the effects, an important control condition would have been a second stimulus pair C-D, where D is not paired with shocks (in contrast to A-B). It would have been useful to see that the animals can discriminate between a shocked (B) and an unshocked (D) stimulus and only freeze in response to B. Likewise, in such a situation one would expect freezing in response to A, but not C. In addition, it would have been useful to see that the response to C is not affected by the various manipulations. While the reviewers agree that an additional control experiment is not strictly necessary, these issues should at least be discussed. For instance, is it possible that the absence of a C-D pair in the current design could explain the apparent dominance of mediated learning?

We take the point that the within-subject design used in other studies has a number of virtues. In the revised manuscript, we discuss the value of this design, inferences supported by the use of this design, and the potential role of the PRh in coding for mediated associations in this design (see below).

Two new paragraphs in the Discussion:

“The present study used a between-subject design to show that freezing to the sensory preconditioned sound is doubly contingent on sound-light and light-shock pairings across the two stages of training; and that the PRh codes a mediated sound-shock association during stage 2 conditioning of the light. It does, however, leave open the question of whether the PRh codes for other types of mediated associations. This question could be assessed using a within-subject design, in which rats are exposed to two sets of stimulus pairings, e.g., tonelight and noise-flashing light (flash), in stage 1; shocked presentations of the light and nonshocked presentations of the flash in stage 2; and test presentations of the tone and noise in stage 3. Sensory preconditioning would be evident as greater test responding to the tone than the noise (e.g., Jones et al., 2012; see also Sharpe et al., 2017). Such a difference could originate in two distinct associations formed across stage 2 of training: an excitatory lightshock association and an inhibitory flash-no shock association. That is, just as the shocked presentations of the light allow the tone to enter into an association with the shock, nonreinforced presentations of the flash should allow the noise to enter into an inhibitory, no shock association. In other words, the test differences between the tone and the noise could be due to the former enhancing and the latter reducing freezing. Since both the tone-light and the noise-flash associations are likely to be encoded in the PRh in stage 1, we predict that silencing activity in the PRh across the discriminative conditioning of the light and flash in stage 2 would eliminate the test differences between the tone and the noise. It would do so by preventing not only the mediated excitatory conditioning of the tone but also the mediated inhibitory conditioning of the noise.”

“It is worth noting that the within-subject design just described requires many trials to achieve differential responding in stage 2; that is, for the rats to freeze when presented with the shocked light and not to freeze when presented with the non-shocked flash. In contrast, relatively few shocked presentations were used here. This difference may be important with respect to when the information acquired in sensory preconditioning and conditioning is integrated. There is evidence that the likelihood of mediated conditioning decreases with the number of conditioning trials (Holland, 1998). Moreover, Jones et al (2012) used just such a within-subject, sensory preconditioning design in an appetitive conditioning protocol (one that involved multiple trials to produce the discrimination in stage 2) to show that integration occurred at the time of testing: specifically, they reported that silencing the orbitofrontal cortex prior to testing reduced the expression of sensory preconditioned appetitive responding (see also Sharpe et al., 2017). Nonetheless, it remains to be shown within a single study that variations in the number of stage 2 conditioning trials regulates whether integration occurs in stage 2 (online) or stage 3 (memory chaining).”

[Editors' note: further revisions were requested prior to acceptance, as described below.]

The authors have adequately addressed essential points 1 and 3 in addition to all minor points. As a result, the manuscript has improved but there are some remaining issues that need to be addressed before acceptance. Specifically, the reviewers felt that the authors did not sufficiently respond to essential point 2, in which we asked for discussion of an alternative interpretation of the current findings.During the initial round of reviews, all reviewers agreed that the current results do not rule out the idea that integration happens during phase 3. They felt that the current findings are also compatible with the idea that integration occurs in phase 3 (i.e., chaining). To recapitulate this point: It is possible that PRh encodes cue-cue associations during phase 1 and cue-outcome associations during phase 2, but that PRh is also critical for integrating cue-cue and cue-outcome associations during phase 3, while another area (e.g. amygdala) additionally stores cue-outcome associations during phase 2. If PRh is inactive during any of these three phases, responding to the indirectly associated cues will be disrupted in phase 3, even if integration happens exclusively in phase 3 (i.e., even if there is no mediated learning at all during phase 2). Specifically, inactivation during phase 2 will prevent encoding of cue-outcome associations which are required for integration in phase 3. In contrast, responding to directly experienced cue-outcomes associations in phase 3 is not impaired by inactivating PRh at any stage because these associations are additionally stored in the amygdala, but they cannot be used for integration. This idea would also explain why PRh inactivation during phase 3 disrupts responding to indirectly associated cues.The authors did not mention this alternative interpretation in the new Discussion paragraph. We would like to ask you to discuss this alternative interpretation of your results and to clearly state that the current results do not rule out the idea that integration happens during phase 3.

We apologise for not adequately addressing the second of the major points in the original review. We have included new text in the Discussion of the further revised manuscript that states the following: “Nonetheless, we acknowledge that even these results do not completely rule out the possibility that some portion of the integration occurs at the time of testing in stage 3. For example, given the initial coding of the light in the PRh in stage 1 (as part of the sound-light association), it is possible that the light-shock association that forms in stage 2 is coded in parallel in the amygdala (principally) and PRh. If so, the PRh may further support responding to the sensory preconditioned sound by integrating the sound-light and lightshock associations at the point of testing (or at least the portions of these associations that are represented in the PRh). According to this account, silencing the PRh during stage 2 would have prevented coding of the light-shock association in this region, thereby undermining the possibility of memory chaining during testing in stage 3; and silencing the PRh during stage 3 would have blocked memory chaining directly, thereby disrupting responding to the preconditioned sound.”

We also made some very minor edits to the final summary for consistency with the new added text: e.g., we changed “showed” to “suggested”.